# Zero-Shot Rankability: Revealing Latent Ordinal Structure in Multimodal Large Language Models via Language

Nam Hyeon-Woo [1]   Moon Ye-Bin [1]   Sohwi Lim [2]   Kwon Byung-Ki [3]   Tae-Hyun Oh [4]

## Abstract

Recent work shows that vision encoders capture ordinal attributes along linear axes, which can be recovered from as few as two labeled images. However, in the zero-shot setting, the text-driven rank axis for Vision-Language Models (VLMs) like CLIP remains suboptimal. In this work, we study the embeddings of Multimodal LLMs (MLLMs). We hypothesize that MLLMs can overcome this limitation due to three potential advantages: their inherent ordinal understanding, capacity for conditional embeddings, and a small cross-modal gap. We show that MLLMs are rankable using only text prompts. Experiments demonstrate that a text-driven rank axis for MLLM embeddings achieves 90% of the performance of the supervised linear rank axis, significantly outperforming the 61% observed in VLM embeddings. We validate that this capability stems from MLLMs' conditional embeddings and a smaller modality gap than VLMs. Furthermore, we demonstrate that this property generalizes to the audio domain. Our findings suggest that language provides a direct interface for probing latent ordinal structures in MLLMs. Code and prompts are available at ○ kaist-ami/prompt-probing.

## 1. Introduction

Large autoregressive models, such as large language models (LLMs) and their multimodal extensions (MLLMs), are inherently generative. Recently, some work has repurposed their internal hidden states as general embeddings (Jiang et al., 2024a; Li & Zhou, 2025; Jiang et al., 2025; 2024b).

[1]Electrical Engineering, POSTECH, Pohang, South Korea [2]School of Electrical Engineering, KAIST, Daejeon, South Korea [3]Grad. School of Artificial Intelligence, POSTECH, Pohang, South Korea [4]School of Computing, KAIST, Daejeon, South Korea. Correspondence to: Tae-Hyun Oh <taehyun.oh@kaist.ac.kr>.

*Proceedings of the 43rd International Conference on Machine Learning*, Seoul, South Korea. PMLR 306, 2026. Copyright 2026 by the author(s).

Unlike encoders trained with contrastive or representation-learning objectives, these embeddings inherit geometric structures induced by next-token prediction. How such training shapes the embedding space, however, remains poorly understood, particularly with respect to which structural properties are preserved and which are lost.

Embeddings can be categorized as similarity embeddings, which preserve local metric distances (Radford et al., 2021), and ordinal embeddings, which preserve the global ordering of attributes (Parikh & Grauman, 2011; Vendrov et al., 2016; Wang et al., 2023b). The magnitude along an ordinal direction corresponds directly to the intensity of the attribute (Fig. 2). We investigate whether MLLM embeddings exhibit such an ordinal structure as a concrete step toward understanding their embedding geometry.

A standard tool for interpreting an embedding space is linear probing (Fig. 1, Left), where a linear layer is trained on top of frozen embeddings to predict a target attribute (Alain & Bengio, 2017; Huang et al., 2024; Kirichenko et al., 2023). Extending this perspective to ordinal attributes, Sonthalia et al. (2025) introduce *rankability*, the property that there exists a linear axis $\mathbf{a}_c$ preserving the ordering of images for a given attribute $c$, namely $\mathbf{h}_i \succeq \mathbf{h}_j \implies \mathbf{a}_c^\top \mathbf{h}_i \geq \mathbf{a}_c^\top \mathbf{h}_j$. They show that modern vision encoders, including ResNet (He et al., 2016) and CLIP (Radford et al., 2021), exhibit rankability, and that $\mathbf{a}_c$ can be recovered from only a few labeled images.

A more natural and practical alternative is to recover $\mathbf{a}_c$ directly from text, because LLMs and MLLMs fundamentally operate through a language interface. Sonthalia et al. (2025) already explore this approach on vision language models (VLMs) and find that the text-driven axis falls well below linear probing, suggesting that it remains suboptimal in the VLM setting. We formalize this capability as a distinct embedding property, *Zero-Shot Rankability*, defined as the SRCC of a rank axis constructed solely from prompts referring to attribute $c$. We refer to the underlying procedure as *prompt probing* (Fig. 1, Right), where the rank axis is inferred directly from language prompts rather than learned through supervised linear probing. This formulation allows us to quantify how much ordinal structure is accessible through language prompts alone.

Prompt probing is particularly appealing in the LLM/MLLM era.[1] Unlike linear probing, which requires labeled images and a separately trained layer for each attribute, prompt probing recovers the rank axis directly from language instead of learning a task-specific linear probe. The procedure also mirrors how MLLMs are naturally used through language interfaces, while revealing an ordinal structure that is directly accessible through prompting rather than learned by a probe. Although prompt probing is inherently more constrained than supervised probing, we argue that it offers a complementary and practically meaningful lens for understanding the geometry of MLLM embeddings.

Applying prompt probing to MLLM embeddings reveals a different trend from prior observations on VLMs. MLLM embeddings are not only linearly rankable, on par with modern vision encoders, but also strongly zero-shot rankable, with the text-driven axis recovering, on average, 90% of linear probing performance across eight vision datasets spanning Age, Crowd, Aesthetics, and Recency. In contrast, VLMs recover only 61% on the same metric, consistent with prior reports.

We further investigate why MLLM embeddings close this zero-shot rankability gap. Removing the attribute-specific instruction prompt drops zero-shot rankability sharply, suggesting that conditional embeddings disentangle the target attribute from polysemantic interference. We then present three lines of evidence identifying the modality gap as the primary obstacle for VLMs: (1) layer-wise rankability in MLLMs rises in step with the shrinking modality gap, (2) a post-hoc gap-reduction transformation on frozen VLMs recovers a large fraction of their zero-shot deficit, and (3) a cross-attention versus projector comparison within the MLLM family reveals the same trend. Together, these findings explain the contrasting behavior between MLLMs and VLMs. Our contributions are summarized as follows:

- We articulate *Zero-Shot Rankability* as a distinct embedding property, and introduce *prompt probing* as a language-driven counterpart to linear probing for analyzing ordinal geometry in MLLM embeddings.

- We show that MLLM embeddings exhibit strong zero-shot rankability, with text-driven axes recovering 90% of linear probing performance across eight vision datasets, revealing an emergent ordinal structure not observed in prior VLM studies.

- We identify the two mechanisms underlying this contrast, namely conditional embeddings and a reduced modality gap, through ablations, layer-wise analyses, and a causal gap-reduction experiment on frozen VLMs.

---

[1]We distinguish VLMs and MLLMs. VLMs do not have a language decoder, such as CLIP. MLLMs are composed of vision encoders and an LLM.

- We extend prompt probing to audio through Omni-MLLMs, uncovering a unified ordinal geometry that supports asymmetric cross-modal axis transfer.

## 2. Related Work

**Visual and Multimodal Embeddings.** Modern vision and multimodal models rely on embeddings as the foundation for downstream perception and reasoning tasks (Bengio et al., 2013; He et al., 2020; Radford et al., 2021; He et al., 2022). Early work primarily focuses on discriminative models, such as ResNet (He et al., 2016), for classification. With the advent of contrastive learning, VLMs (Radford et al., 2021) are trained to align visual and textual representations in a shared metric space. This alignment enables zero-shot capabilities. These embeddings have been widely used and analyzed for their similarity structure (Gordo et al., 2016; Kordopatis-Zilos et al., 2025; Lim et al., 2026) and geometric properties (Lewis et al., 2024; Fahim et al., 2024; Huh et al., 2024; Berasi et al., 2025; Ye-Bin et al., 2023). For instance, Liang et al. (2022) have analyzed the modality gap, a phenomenon where visual and textual embeddings occupy distinct regions in the shared space. Berasi et al. (2025) have investigated the compositionality of visual embedding. Mistretta et al. (2025) have introduced the notion of intra-modal misalignment, which leads to suboptimal performance on intra-modal tasks. While these works provide deep insights into the clustering and separability of embeddings, they largely focus on semantic similarity rather than continuous ordinal relationships.

**Ordinality and Rankability in Embeddings.** Ordinal embeddings aim to capture continuous or ordered attributes (*e.g.*, age, aesthetic score) from data (Parikh & Grauman, 2011; Vendrov et al., 2016). Traditional approaches typically rely on specialized ranking losses or architectures (Liu et al., 2018; 2019; Li et al., 2021; Ke et al., 2021). More recently, there has been growing interest in extracting ordinal information from pre-trained vision-language models, either by fine-tuning CLIP with ordinal supervision or by aligning images with ordered text descriptions through prompt engineering (Wang et al., 2023b;a; Li et al., 2022; Yu et al., 2025; Du et al., 2024).

Recently, Sonthalia et al. (2025) introduced the concept of *rankability*, showing through linear probing (Alain & Bengio, 2017; Huang et al., 2024; Kirichenko et al., 2023) that modern visual embeddings inherently encode continuous attributes along a linear *rank axis*. They demonstrate that a simple linear projection can recover the global ordering of attributes such as age and crowd count. However, their analysis is limited to conventional visual encoders (*e.g.*, ViT, ResNet, CLIP). In contrast, we study rankability in MLLM embeddings, which operate in a distinct generative

*Figure 1.* **Linear probing versus prompt probing. Left:** Linear probing trains a linear layer on frozen embeddings to predict an attribute. **Right:** Prompt probing constructs the rank axis $\mathbf{a}_c$ directly from language prompts using prompts, revealing latent structure via language.

embedding space, and introduce *prompt probing*, a language-driven counterpart to linear probing that recovers the rank axis directly from text.

**Embeddings of MLLMs** MLLMs (Liu et al., 2023; Bai et al., 2025b;a; Achiam et al., 2023) integrate visual encoders with LLMs (Brown et al., 2020; Raffel et al., 2020; Yang et al., 2025; Guo et al., 2025) to perform complex reasoning and generation tasks. Generative embedding of LLMs has emerged (Jiang et al., 2024a; Li & Zhou, 2025), which inspires the embedding of MLLMs (Jiang et al., 2025; 2024b; Kim et al., 2026). Specifically, to extract the embedding from LLMs or MLLMs, a prompt such as *"This sentence or image means in one token:"* is given and the last-layer token before the decoding head is used as the embedding vector. A parallel line of work probes the internal geometry of LLM hidden states through linear probes (Alain & Bengio, 2017) and the logit lens (nostalgebraist, 2020), yet the ordinal structure of MLLM embeddings remains underexplored.

# 3. Background and Probing Setup

**Notation.** We denote an image dataset by $\mathcal{X}$ and an ordinal attribute by a function $A : \mathcal{X} \to \mathcal{Y} \subset \mathbb{R}$, where $A(x)$ is the ground-truth label of $x$. An embedding function $f$ maps its input(s) to a vector $\mathbf{h} \in \mathbb{R}^d$; for MLLMs, $f$ may take an instruction prompt together with a content input (text or image). Throughout, $f$ is taken to return $\ell_2$-normalized vectors. We use $\rho$ for Spearman's rank correlation coefficient (SRCC).

## 3.1. Background

**Rankability.** Sonthalia et al. (2025) introduce *rankability* as a linear-probing property of embeddings.

**Definition 3.1** (Rankability). An embedding function $f$ is *rankable* for an ordinal attribute $A$ over $\mathcal{X}$ if there exists a *rank axis* $\mathbf{a}_c \in \mathcal{S}^{d-1}$ such that for any $x_1, x_2 \in \mathcal{X}$ with $A(x_1) \geq A(x_2)$, we have $\mathbf{a}_c^\top f(x_1) \geq \mathbf{a}_c^\top f(x_2)$.

In practice, $\mathbf{a}_c$ is estimated by supervised linear regression on $\{(f(x_i), A(x_i))\}_i$ and rankability is measured as $\rho$ on a

held-out test set. They show that modern visual embeddings (ResNet (He et al., 2016), CLIP (Radford et al., 2021)) are rankable, and propose an efficient method that recovers $\mathbf{a}_c$ from a few labeled images. They also explore a zero-shot text-based approach for VLMs, but the resulting rank axis falls well below the image-based linear probe.

**MLLM Embedding.** LLMs and MLLMs operate autoregressively, and recent work extracts embeddings from their internal hidden states (Jiang et al., 2024a; Li & Zhou, 2025; Jiang et al., 2025; 2024b). We follow the Prompt with Explicit One Word Limitation method (Jiang et al., 2024a). Given an instruction prompt $\mathbf{p}$ (e.g., *"This [input] means in one token:"*) and an input $x$, we compute the hidden states $[\mathbf{h}_n^l]_{l,n} = \text{MLLM}(\mathbf{p}, x)$, where $l$ indexes layer and $n$ indexes token. The embedding is the last-token hidden state of the final layer, $\mathbf{h} \equiv \mathbf{h}_N^L = f(\mathbf{p}, x)$.

## 3.2. Probing Setup

**Zero-Shot Rankability.** Sonthalia et al. (2025) already measure SRCC for VLMs by constructing $\mathbf{a}_c$ from prompts rather than from labels, and observe that this text-driven score falls well below the supervised rankability upper bound. While the measurement appears in prior work, it has not been articulated as a distinct property of embeddings. We formalize it as *Zero-Shot Rankability*.

**Definition 3.2** (Zero-Shot Rankability). For an embedding function $f$ and a target ordinal attribute $A$, *zero-shot rankability* is the SRCC $\rho$ between $\{\mathbf{a}_c^\top f(x_i)\}_i$ and $\{A(x_i)\}_i$ on a held-out test set, where $\mathbf{a}_c$ is constructed from text prompts referring to attribute $c$ but containing no labeled image–label examples.

Here *zero-shot* follows the standard LLM/MLLM setting of using no in-context training examples in the prompt. The target attribute itself (*e.g.*, age or aesthetics) is specified in language, but no labeled $(x, A(x))$ pairs are included to construct the rank axis.

For VLMs, Sonthalia et al. (2025) report that the text-driven axis falls well below supervised rankability, leaving zero-shot rankability limited. However, MLLMs exhibit strong

ordinal capabilities in generation, both for pairwise comparison and absolute estimation. For example, they achieve pairwise "who is older" accuracy above $0.83$ when comparing two images, while age predictions from a single image reach an SRCC of $0.772$ with ground truth (UTKFace, Qwen2.5-VL-7B). These results indicate that the ordinal knowledge required to rank images by $c$ is already present inside the MLLM, since the model can verbalize a calibrated estimate directly from pixels. Whether this knowledge is also linearly accessible from the embedding through a text-driven axis is the central empirical question.

**Probing Protocol: Text-Driven Rank Axis.** The protocol uses language inputs in two roles (Fig. 1). First, a pair of *antonymous anchor texts* $T_c^{\text{pos}}, T_c^{\text{neg}}$ describes the two ends of the attribute axis. The rank axis is the difference of their text embeddings,

$$\mathbf{a}_c = \frac{f(T_c^{\text{pos}}) - f(T_c^{\text{neg}})}{\|f(T_c^{\text{pos}}) - f(T_c^{\text{neg}})\|_2}. \tag{1}$$

For VLMs, $f(T)$ is the output of the language encoder. For MLLMs, $f(T) = f(p_{\text{w}}, T)$ uses a one-word wrapper $p_{\text{w}}$ (Jiang et al., 2024a) as in our MLLM embedding extraction.

Second, for MLLMs an *attribute-specific instruction prompt* $P_c$ conditions the image embedding, giving $\mathbf{h}_x = f(P_c, x)$. For VLMs, conditional image embedding is not available, so $\mathbf{h}_x = f(x)$ is the standard image-encoder output. The zero-shot rank score for $x$ is $s_x = \mathbf{a}_c^\top \mathbf{h}_x$. We give a concrete example for the age attribute below.

$$
\begin{aligned}
T_c^{\text{pos}} &= \text{"A portrait of an aged person"}, \\
T_c^{\text{neg}} &= \text{"A portrait of a young person"}, \\
p_{\text{w}} &= \text{"Describe this sentence in one word:"}, \\
P_c &= \text{"Describe this person's age in one word:"}.
\end{aligned}
$$

*Prompt probing.* The rank axis $\mathbf{a}_c$ depends only on the anchor texts $(T_c^{\text{pos}}, T_c^{\text{neg}})$, and the conditional image embedding $\mathbf{h}_x$ depends only on $P_c$. We search over candidate prompts on the validation set. The anchor texts $(T_c^{\text{pos}}, T_c^{\text{neg}})$ are searched for both VLMs and MLLMs, and additionally $P_c$ for MLLMs.

This formalization is particularly relevant in the MLLM era. Linear probing tunes weight parameters on labels to expose the geometry of an embedding space, but in MLLMs the prompt itself is what shapes the embedding, so the natural object to tune for analysis is the prompt rather than a learned probe.

*Table 1.* **Rankability of VMs and MLLMs.** SRCC are averaged over seven VMs and four MLLMs, respectively. Furthermore, these values represent an average across the datasets corresponding to each attribute. The results indicate that the rankability of MLLM embeddings is comparable to that of VM embeddings.

| Embedding | Age | Crowd | Aesthetics | Recency | Average |
|---|---|---|---|---|---|
| VM | 0.803 | 0.793 | 0.704 | 0.632 | 0.753 |
| MLLM | 0.867 | 0.872 | 0.733 | 0.688 | 0.813 |

## 4. Are MLLM Embeddings Zero-Shot Rankable?

### 4.1. Experimental setup

**Datasets.** We use eight datasets spanning four attribute types. For **Age**, we use UTKFace (Zhang et al., 2017) and Adience (Eidinger et al., 2014). For **Crowd**, UCF-QNRF (Idrees et al., 2018), ShanghaiTech-A, and ShanghaiTech-B (Zhang et al., 2016). For **Aesthetics**, AVA (Murray et al., 2012) and KonIQ-10k (Hosu et al., 2020). For **Recency**, HCI (Palermo et al., 2012).

**Models.** We compare three families of encoders. **Vision-only** models include ResNet-50 (He et al., 2016), ViT-B/32 (Dosovitskiy et al., 2021), ConvNeXt (Woo et al., 2023), and DINOv2-B/14 (Oquab et al., 2024). **VLMs** include three CLIP variants (Radford et al., 2021), namely CLIP-RN50, CLIP-ViTB/32, and CLIP-CNX. **MLLMs** include Qwen 2.5-VL 7B (Bai et al., 2025b) and Qwen 3-VL 2B/4B/8B (Bai et al., 2025a).

**Linear probing.** For rankability, $\mathbf{a}_c$ is fit by linear regression on frozen embeddings. Training, validation, and test sets are strictly separated. Hyperparameters are selected by validation SRCC and the test set is evaluated only once. When no official validation split exists, we hold out 10% of the training set with a fixed random seed (*e.g.*, UTK-Face yields $11,831/1,315/3,287$ train/val/test images; per-dataset sizes are listed in Appendix A.1). We grid-search batch size $\{32, 64, 128\}$, learning rate $\{10^{-1}, \ldots, 10^{-6}\}$, and weight decay $\{10^{-3}, \ldots, 10^{-6}\}$.

**Prompt probing.** We run prompt search on the validation set. We search over 100 GPT-generated antonymous anchor-text pairs $(T_c^{\text{pos}}, T_c^{\text{neg}})$ for both VLMs and MLLMs, and additionally over 100 attribute-specific instruction prompts $P_c$ for MLLMs. The selected prompts are evaluated once on the test set, and no labeled image is included for the axis construction. The prompt list is released with our code (🞉).

### 4.2. MLLM embeddings are linearly rankable

Before turning to the zero-shot setting, we first verify that MLLM embeddings, like modern visual embeddings, are

rankable in the linear-probing sense. Table 1 reports the average SRCC across the four attribute types. MLLMs achieve an average SRCC of 0.813, comparable to the 0.753 achieved by the seven non-MLLM encoders (vision-only models and VLMs), with consistently strong performance across attributes. These results indicate that a linear ordinal axis $\mathbf{a}_c$ exists in the MLLM embedding space.

### 4.3. MLLM embeddings are Zero-Shot rankable

Having established that the rank axis $\mathbf{a}_c$ exists in the MLLM embedding space, we next ask whether it can be recovered from text alone. Table 2 compares the zero-shot rankability of MLLM and VLM embeddings against their supervised rankability upper bounds. For VLM embeddings, the average zero-shot SRCC is 0.497, recovering only 61% of the 0.820 upper bound, consistent with the suboptimality reported (Sonthalia et al., 2025). In contrast, MLLM embeddings achieve an average zero-shot SRCC of 0.728, recovering 90% of the 0.813 upper bound. The improvement is consistent across attributes, with especially large gains on the Aesthetics and Crowd benchmarks. Overall, these results suggest that the previously observed limitation of zero-shot rankability is specific to VLMs and is mitigated in MLLM embeddings.

**Inference cost.** Per-image MLLM embedding extraction is several times more expensive than VLM extraction, requiring 52.76 ms for Qwen 3 VL 2B compared to 13.92 ms for CLIP-CNX on a single A6000 GPU. Within MLLMs, however, embedding extraction remains substantially cheaper than generation-based scoring, taking 92 ms per image compared to 152–1600 ms depending on the decoding length. Despite this efficiency gap, the two approaches achieve comparable performance: embedding-based ranking reaches an SRCC of 0.783 on UTKFace, while generation-based scoring achieves 0.772 on the same model. Prompt probing therefore preserves the ordinal capability of MLLM embeddings at a much lower computational cost, supporting its use as a practical tool for studying embedding geometry.

### 4.4. Qualitative analysis

Figure 2 presents images sorted by their predicted rank scores, together with the corresponding ground-truth labels. Across all attributes, the predicted ordering closely follows the underlying ordinal trend. For Crowd, the ranking aligns with monotonically increasing crowd counts, reflecting a clear visual notion of magnitude. For Age, the overall progression from younger to older subjects is well preserved, with occasional local inversions between visually similar ages. For Aesthetics, low- and high-quality images are clearly separated, while mid-range examples exhibit less consistent ordering, reflecting the subjective nature of aesthetic evaluation. Overall, the text-driven rank axis suc-

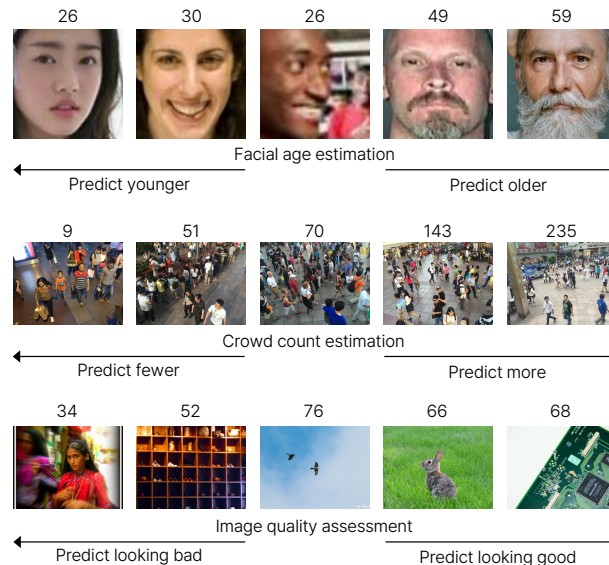

*Figure 2.* **Qualitative visualization of discovered rank axes from text.** We display images sorted by the rank scores predicted by our method (from left to right: 0th to 100th percentiles). The above numbers indicate the ground-truth labels. The model successfully aligns images according to their ordinal information, with minor inversions in subjective or ambiguous cases.

cessfully recovers the global ordinal structure across diverse attributes.

## 5. Why Does Zero-Shot Rankability Emerge?

### 5.1. Conditional embeddings disentangle ordinal attributes

A defining feature of MLLM embeddings, absent in VM and VLM encoders with fixed visual features, is that the embedding $\mathbf{h} = f(\mathrm{p}, \mathbf{x})$ depends on the text prompt p. On the ColoredMNIST benchmark (Bahng et al., 2020) (10 digits, 10 colors), the prompt *"Summarize the color of the image in one word:"* returns embeddings whose top-10 nearest neighbors cluster by color (Fig. 3). The same image admits different conditional embeddings that isolate different attributes. We hypothesize that this conditioning is what enables zero-shot ranking; the conditional image prompt $P_c$ aligns the image embedding with attribute $c$, so the text-derived axis $\mathbf{a}_c$ projects onto a representation already focused on $c$.

**Ablation on the conditional prompt.** We test this hypothesis by replacing the attribute-specific $P_c$ with a generic prompt (*e.g.*, *"Summarize this image in one word:"*). Table 3 reports zero-shot rankability with and without conditional prompting across four MLLMs. Removing the conditional prompt reduces the average SRCC of Qwen 2.5 VL 7B from 0.734 to 0.544, with a similar trend observed across the other

*Table 2.* **MLLM embeddings are Zero-Shot rankable.** We report SRCC. **Rankability** is the supervised linear-probing upper bound. **Zero-Shot Rankability** is the SRCC of a text-driven rank axis. MLLMs recover 90% of the rankability upper bound, while VLMs recover only 61%. † denotes the value reported in Sonthalia et al. (2025).

| Embedding | Metric | Age | | Crowd | | | Aesthetics | | Recency | Average |
|---|---|---|---|---|---|---|---|---|---|---|
| | | UTKFace | Adience | QNRF | ST-A | ST-B | AVA | KonIQ | HCI | |
| VLM | Rankability | 0.804 | 0.910 | 0.872 | 0.764 | 0.878 | 0.718 | 0.838 | 0.778 | 0.820 |
| | Zero-Shot Rankability† | 0.600 | 0.782 | 0.315 | 0.242 | 0.535 | 0.410 | 0.547 | 0.449 | 0.485 (59%) |
| | Zero-Shot Rankability | 0.700 | 0.815 | 0.555 | 0.564 | 0.677 | 0.428 | 0.499 | 0.423 | 0.497 (61%) |
| MLLM | Rankability | 0.816 | 0.919 | 0.879 | 0.825 | 0.911 | 0.699 | 0.768 | 0.688 | 0.813 |
| | Zero-Shot Rankability | 0.797 | 0.890 | 0.768 | 0.619 | 0.892 | 0.599 | 0.774 | 0.485 | 0.728 (90%) |

*Table 3.* **Conditional embeddings improve Zero-Shot Rankability.** We measure SRCC with and without conditional embeddings. Conditional embedding improves SRCC.

| Model | Conditional Prompt | Age | | Crowd | | | Aesthetics | | Recency | Average |
|---|---|---|---|---|---|---|---|---|---|---|
| | | UTKFace | Adience | QNRF | ST-A | ST-B | AVA | KonIQ | HCI | |
| Qwen 2.5 VL 7B | ✗ | 0.720 | 0.848 | 0.579 | 0.634 | 0.724 | 0.260 | 0.349 | 0.234 | 0.544 |
| | ✓ | 0.783 | 0.880 | 0.756 | 0.636 | 0.880 | 0.600 | 0.763 | 0.571 | **0.734** |
| Qwen 3.0 VL 8B | ✗ | 0.749 | 0.856 | 0.573 | 0.677 | 0.815 | 0.316 | 0.412 | 0.255 | 0.558 |
| | ✓ | 0.804 | 0.888 | 0.747 | 0.509 | 0.875 | 0.613 | 0.778 | 0.465 | **0.710** |
| Qwen 3.0 VL 4B | ✗ | 0.762 | 0.880 | 0.695 | 0.663 | 0.759 | 0.293 | 0.412 | 0.231 | 0.587 |
| | ✓ | 0.800 | 0.885 | 0.812 | 0.655 | 0.928 | 0.597 | 0.786 | 0.484 | **0.743** |
| Qwen 3.0 VL 2B | ✗ | 0.762 | 0.886 | 0.597 | 0.503 | 0.788 | 0.296 | 0.454 | 0.062 | 0.544 |
| | ✓ | 0.800 | 0.908 | 0.758 | 0.675 | 0.884 | 0.584 | 0.768 | 0.418 | **0.724** |

Query | Top 10 Retrieved Images

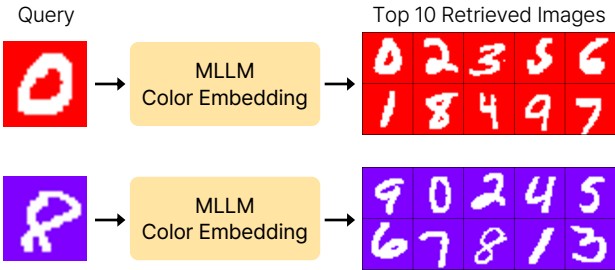

*Figure 3.* **Conditional embeddings disentangle attributes on ColoredMNIST.** Top-10 nearest neighbors of MLLM embeddings extracted with attribute-specific conditional prompts.

three models. The degradation is particularly pronounced for more abstract attributes: performance on Aesthetics (KonIQ) drops from 0.763 to 0.349, and Recency (HCI) from 0.571 to 0.234, whereas more concrete attributes such as Age are affected considerably less.

**Conditional prompting also improves the supervised upper bound.** An additional question is whether the conditional prompt merely helps identify the rank axis or also improves the embedding itself. Table 4 reports the linearly probed rankability, *i.e.*, the supervised upper bound, with and without conditional prompting. Conditional prompting increases the average SRCC from 0.799 to 0.839. As before, concrete attributes show only modest improvements, while

*Table 4.* **Rankability of the conditional MLLM embeddings.** We measure SRCC with and without conditional embeddings of Qwen 2.5 VL 7B. The improvement in rankability indicates that conditional embeddings help disentangle ordinal features.

| Conditional Prompt | Age | Crowd | Aesthetics | Recency | Average |
|---|---|---|---|---|---|
| ✗ | 0.858 | 0.866 | 0.697 | 0.683 | 0.799 |
| ✓ | 0.863 | 0.877 | 0.794 | 0.765 | 0.839 |

abstract attributes benefit substantially. Since the supervised probe is trained with the same labels in both settings, the improvement must arise from the embedding itself. These results suggest that conditional prompting reorganizes the representation such that the ordinal direction becomes more linearly recoverable.

**Robustness to the antonymous pair.** In this paper, prompt probing is used as a geometry analysis tool. Like linear probing, it measures what language can read off the embedding rather than what supervision can. Validation-set prompt selection follows directly from that goal. Distinct from this analysis stance, a separate line of work discusses what constitutes *true zero-shot* when prompts are tuned on a validation set (Perez et al., 2021). Our paper does not claim a position in that paradigm, but we report a related check. Holding the rest of the protocol fixed and varying only the antonymous pair, we report the mean and standard deviation of SRCC without selection. Qwen 2.5 VL 7B yields

$0.728 \pm 0.061$ on UTKFace and $0.409 \pm 0.182$ on KonIQ. CLIP-CNX yields $0.555 \pm 0.168$ and $0.157 \pm 0.170$ on the same datasets. The MLLM mean stays well above the VLM mean on both datasets, suggesting that the rankability gap is not an artifact of anchor-pair selection.

**Interpretation via polysemanticity.** We interpret this phenomenon through the lens of polysemanticity (Elhage et al., 2022; Gao et al., 2025). Embeddings are typically polysemantic, meaning that a vector $\mathbf{h} \in \mathbb{R}^n$ simultaneously encodes multiple concepts. We decompose $\mathbf{h}$ into a target concept and interfering components:

$$\mathbf{h} = \sum_i w_i \mathbf{v}_i = w_{\text{target}} \mathbf{v}_{\text{target}} + \sum_{j \neq \text{target}} w_j \mathbf{v}_j. \quad (2)$$

Given a candidate axis $\mathbf{a}$, the resulting score is $s = \mathbf{a}^\top \mathbf{h}$. Ideally, $\mathbf{a}$ aligns only with $\mathbf{v}_{\text{target}}$, but in practice high-dimensional concepts are not orthogonal, and the interference term $\mathbf{a}^\top \sum_{j \neq \text{target}} w_j \mathbf{v}_j$ can distort the ranking. We hypothesize that the conditional prompt acts as a semantic filter that amplifies $w_{\text{target}}$ while suppressing irrelevant components $w_j$, thereby reducing interference and making the ordinal direction more recoverable. This effect is large for abstract attributes such as aesthetics, where semantic overlap with unrelated visual features is substantial.

## 5.2. Reduced modality gap aligns text and visual axes

Even when conditional prompting successfully isolates the target attribute, the rank axis $\mathbf{a}_c$ is still derived from text and applied to image embeddings. As a result, any misalignment between the language and visual subspaces can distort the projection, regardless of how cleanly the attribute is represented. Standard VLMs trained with contrastive objectives exhibit a modality gap, where image and text embeddings occupy distinct regions of the shared embedding space (Liang et al., 2022). In contrast, MLLMs trained with shared decoder parameters and autoregressive objectives exhibit better alignment. As shown in Fig. 4, UMAP visualizations of COCO (Lin et al., 2014) image and caption embeddings reveal that Qwen-VL (Bai et al., 2025b) produces overlapping image and text distributions at the final layer, whereas CLIP maintains a clear separation between the two modalities.

To quantify this gap across layers, we compute per-layer vocabulary distributions from the last-token hidden states using the Logit Lens (nostalgebraist, 2020), obtained by applying the model's final unembedding matrix followed by a softmax. We then measure the Jensen–Shannon divergence between the image- and text- distributions. The divergence decreases monotonically with depth, indicating that the modality gap progressively narrows in deeper layers.

**Rankability increases as the modality gap narrows.** If reducing the modality gap improves cross-modal projection, zero-shot rankability should increase in deeper MLLM layers where the gap becomes smaller. Figure 5 reports the zero-shot SRCC obtained by applying the text-driven rank axis to embeddings extracted from each layer of Qwen 2.5 VL 7B. Across all datasets, deeper layers consistently achieve higher SRCC, with the most pronounced improvements appearing in the final layers for Age-related datasets such as UTKFace and Adience. This trend closely follows the corresponding JSD curve of the same model, supporting the hypothesis that reducing the modality gap enables the text-derived axis to align more effectively with the underlying visual ordinal structure.

**Closing the modality gap partially restores zero-shot rankability.** The layer-wise trend alone does not establish causality. To isolate the role of the modality gap, we apply the gap-reduction transformation of I0T (An et al., 2025) to frozen VLM embeddings and re-evaluate zero-shot probing on KonIQ. Zero-shot SRCC improves substantially across all three CLIP variants: CLIP-RN50 improves from $0.466$ to $0.661$, CLIP-ViTB/32 from $0.502$ to $0.663$, and CLIP-CNX from $0.508$ to $0.738$. Since the underlying VLMs remain otherwise unchanged, these results suggest that the modality gap itself is a primary factor limiting text-driven ranking in VLM embeddings.

**Cross-attention versus projector-based MLLMs.** We further examine the same hypothesis through an architectural comparison within the MLLM family. IDEFICS1 (Laurençon et al., 2023) (9B) processes visual inputs through cross-attention layers, maintaining partially separated visual and textual representations, whereas IDEFICS2 (Laurençon et al., 2024) (8B) replaces the cross-attention module with a projector that maps visual tokens directly into the language-model embedding space, resulting in tighter cross-modal alignment. The two models achieve similar performance on UTKFace ($0.774$ vs. $0.751$), where the ordinal signal is relatively concrete, but differ substantially on KonIQ ($0.338$ vs. $0.718$), where successful ranking depends on fine-grained alignment between language and abstract visual attributes. These results further support the hypothesis that tighter modality alignment is particularly important for text-driven ranking of abstract attributes.

**Geometric interpretation.** Let $\mathbf{a}_I$ denote the ideal rank axis in the visual modality and $\mathbf{a}_L$ the axis derived from text. A modality gap induces an angular offset between

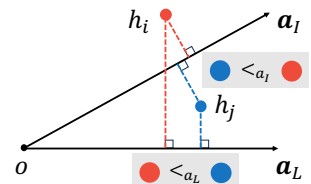

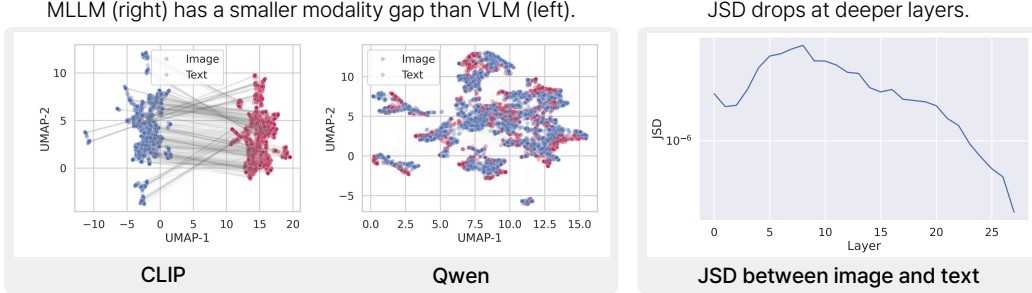

*Figure 4.* **MLLMs close the modality gap.** On COCO, UMAP of image and text embeddings shows overlap at the last layer of Qwen-VL but separation in CLIP. The Logit-Lens Jensen–Shannon divergence between image and text token distributions falls with layer depth in MLLMs.

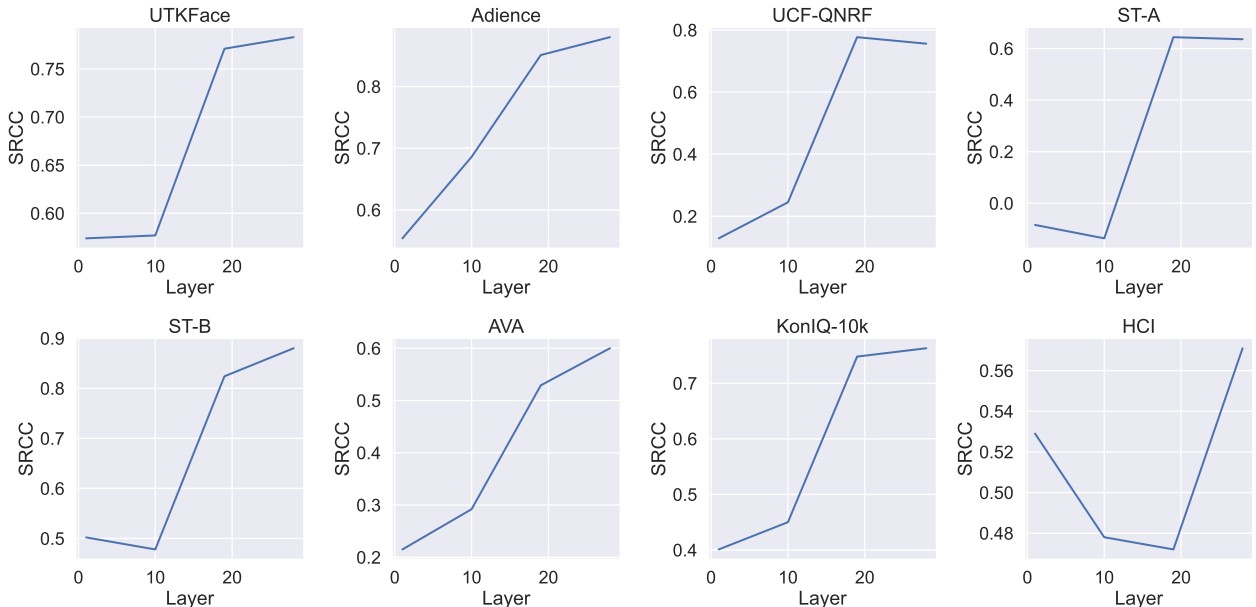

*Figure 5.* **Layer-wise rankability of MLLM embeddings.** We measure SRCC from different layers. The results consistently show that deeper layers achieve significantly higher rankability scores than shallower layers, indicating that ordinal information is more effectively encoded in later stages, where the modality gap is reduced.

$\mathbf{a}_I$ and $\mathbf{a}_L$, and projecting onto a misaligned axis can flip the order of two points whose visual scores are close,

$$\mathbf{a}_I^\top \mathbf{h}_i < \mathbf{a}_I^\top \mathbf{h}_j \quad \text{but} \quad \mathbf{a}_L^\top \mathbf{h}_i > \mathbf{a}_L^\top \mathbf{h}_j. \quad (3)$$

Larger angles produce more frequent inversions and thus lower SRCC. Reducing the gap brings $\mathbf{a}_L$ closer to $\mathbf{a}_I$, preserving the intrinsic ordinal inequality during cross-modal projection.

## 6. Cross-Modal Ordinal Geometry

Omni MLLMs accept audio inputs through the same prompt-conditioned interface used for vision, so the probing protocol of Sec. 4 applies to audio embeddings without modification. We test this on the age attribute with Common Voice (Ardila et al., 2020) and Qwen 2.5 Omni 7B.

Both the audio-conditional prompt $P_c$ and the anchor texts $(T_c^{\text{pos}}, T_c^{\text{neg}})$ are searched on the validation set following the same protocol as in vision; one representative $P_c$ is *"Identify the age group of this person in one word."* Sec. 6.1 reports rankability on audio alone, and Sec. 6.2 examines whether text-driven rank axes transfer between vision and audio.

### 6.1. Audio embeddings are rankable

On Common Voice, Qwen 2.5 Omni 7B reaches a supervised rankability of $0.468$ (Table 5), substantially lower than on facial age (UTKFace, Adience). This may simply reflect that inferring age from voice alone is more difficult than from facial appearance. Within this lower ceiling, the text-driven rank axis still recovers an SRCC of $0.409$, or $87\%$ of the supervised upper bound, on par with the $90\%$ recovery

*Table 5.* **Rankability of Omni MLLM Embedding in the sound modality.** We measure SRCC with Qwen 2.5 Omni 7B on the Common Voice dataset. Despite lower rankability, omni MLLM embedding achieves 87% zero-shot rankability compared to linear-probing rankability.

|  | Zero-Shot (Ours) | Rankability |
|---|---|---|
| Qwen 2.5 Omni 7B | 0.409 (87%) | 0.468 |

*Table 6.* **Cross-modal transferability of rank axes.** Rows represent the Source dataset used to discover the rank axis, while columns represent the Target dataset for evaluation. We report the performance ratio relative to the self-reference baseline, where performance is measured by SRCC reported in parentheses. V and A denote Vision and Audio modalities, respectively. The results indicate that V-to-V and A-to-V transfers are highly effective; however, V-to-A transfer exhibits significant performance degradation.

| **Modality:** | **Target** | | |
|---|---|---|---|
| A (Audio), V (Image) | Common Voice (A) | UTKFace (V) | Adience (V) |
| **Source** Common Voice (A) | 1.0 (0.409) | 0.96 (0.775) | 0.73 (0.647) |
| UTKFace (V) | 0.53 (0.216) | 1.0 (0.783) | 0.93 (0.815) |
| Adience (V) | 0.30 (0.123) | 0.93 (0.727) | 1.0 (0.880) |

on vision (Sec. 4). The same protocol thus tracks the available ordinal signal in audio as well as in vision, even though that signal itself is smaller.

### 6.2. Text-driven axes transfer asymmetrically across modalities

Linear probing produces a trained weight vector, while prompt probing produces a set of selected prompts. We test whether these prompts still recover the ordinal axis when applied to a different dataset, even across modalities. The prompts selected on the source dataset are reused on the target dataset. We instantiate this for the age attribute across Common Voice, UTKFace, and Adience. Table 6 reports SRCC together with the ratio against the self-source baseline.

Within vision, axes transfer well, with both directions between UTKFace and Adience reaching 93% of the self-source SRCC. Audio-to-vision transfer also holds, recovering 73% to 96% across the two vision datasets. Vision-to-audio is much weaker, at 30% to 53%. The pattern is asymmetric, with audio-derived axes generalizing to vision more readily than vision-derived axes to audio. A plausible explanation is simply that the age signal is intrinsically stronger in vision embeddings than in audio embeddings. Any reasonable axis projects well onto vision, while transfer to audio is bounded by the weaker age signal in the audio embedding itself.

## 7. Limitations

Prompt probing fundamentally relies on language being able to specify the two ends of a target attribute. As a result, attributes whose extremes are difficult to express through natural antonymous descriptions are inherently harder to recover. This includes attributes such as historical periods (HCI Recency), domain-specific medical or geological scales, and categorical concepts without clear linguistic opposites. The relatively lower zero-shot performance on HCI Recency reflects this limitation.

Another limitation is the possibility of data contamination. Since both VLMs and MLLMs are trained on large-scale web corpora, overlap with the evaluation datasets cannot be completely ruled out. Nevertheless, contamination alone is unlikely to explain our findings, as VLMs and MLLMs are trained on large-scale data sources yet exhibit substantially different levels of zero-shot rankability.

## 8. Conclusion

We formalize *Zero-Shot Rankability* as a distinct embedding property and employ *prompt probing* as a language-driven counterpart to linear probing for the MLLM era, where the rank axis is recovered directly from text. Across eight vision datasets, prompt probing recovers, on average, 90% of the supervised rankability upper bound for MLLM embeddings, whereas the same protocol achieves only 61% on VLM embeddings, consistent with prior observations of VLM suboptimality.

Our analysis attributes this gap to two key mechanisms: conditional embeddings, which help disentangle the target attribute from polysemantic interference, and a reduced modality gap, which better aligns language-derived axes with the underlying visual ordinal structure. We further show that the same framework extends to audio through Omni MLLMs. Taken together, our findings suggest that the latent ordinal structure of MLLM embeddings is directly accessible through language, motivating further study of the emergent geometric properties of MLLM embeddings.

## Acknowledgements

This work was supported by the InnoCORE program of the Ministry of Science and ICT (N10250156); and Institute of Information & Communications Technology Planning & Evaluation (IITP) grants funded by the Korea government (MSIT) (No. RS-2026-25518317, Development of AI memory mechanism that reflects human cognitive principles; No. RS-2024-00457882, National AI Research Lab Project; No. 2022-0-00124, No. RS-2022-II220124, Development of Artificial Intelligence Technology for Self-Improving Competency-Aware Learning Capabilities).

## Impact Statement

This paper studies the geometric properties of pretrained MLLM embeddings through language-driven probing. Although our experiments use publicly available datasets, the same techniques could potentially be applied to sensitive attributes. We encourage future applications to consider standard practices for bias evaluation and responsible deployment.

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

## Contents

# A. Setup

For the dataset and models, we follow the original rankability paper (Sonthalia et al., 2025).

## A.1. Datasets

We adopt a total of eight datasets to comprehensively evaluate the rankability of VLMs and MLLMs.

- **UTKFace:** UTKFace (Zhang et al., 2017) is the age prediction dataset. It contains 13,146 training images and 3,287 test images.

- **Adience** Adience (Eidinger et al., 2014) is the age prediction dataset. It contains 14k images for training, and 4k images for the test set.

- **UCF-QNRF** UCF-QNRF (Idrees et al., 2018) includes images with crowd count annotations in a single image. It consists of 1201 images for training and 334 for the test set.

- **ShanghaiTech-A and ShanghaiTech-B** ST-A/B (Zhang et al., 2016) includes images with crowd count annotations in a single image. Datasets consist of 300 and 400 images for the training, 182 and 316 images for the test set.

- **AVA** AVA (Murray et al., 2012) includes aesthetic annotations in the form of Mean Opinion Scores (MOS) from multiple judgments, with aesthetic quality scores from 1 to 10.

- **KonIQ-10k** Similar to AVA, KonIQ-10k (Hosu et al., 2020) provides image quality Mean Opinion Scores (MOS) collected from multiple crowdsourced ratings on a scale of 1 to 100.

- **HCI** Historical Color Images (Palermo et al., 2012), used here as the **Recency** attribute, contains the time period when photographs were taken. It provides 1,060 images for the training and 265 images for the test set, with 5 ordered decade labels from the 1930s to the 1970s.

- **Common Voice** Common Voice (Ardila et al., 2020) provides various multilingual collections with age annotations for speakers ranging from teens to the eighties. We use an English subset from Common Voice. Age labels are categorical (teens, twenties, ..., eighties); SRCC is computed by comparing the ranks induced by these ordered categories, which is valid for both categorical and continuous ordinal labels.

## A.2. Models

- **ResNet50, ViT-B/32, ConvNeXt, DINOv2 (ViT-B/14):** We use Timm library. For ResNet50 (He et al., 2016), we use "timm_resnet50"; for ViT-B/32 (Dosovitskiy et al., 2021), we use "timm_vit_base_patch32_224"; for ConvNeXt (Woo et al., 2023) we use "timm_convnextv2_large"; and for DINOv2 (Oquab et al., 2024), we use "timm_vit_base_patch14_dinov2".

- **CLIP (ResNet50, ViT-B/32 and ConvNeXt):** We use open clip (Radford et al., 2021; Cherti et al., 2023; Schuhmann et al., 2022). For ResNet50, we use "openai_clip_rn50"; for ViT-B/32, we use "openai_clip_vit_b_32"; and for ConvNeXt, we use "open_clip_convnext_large_d_320".

- **MLLM:** We use Hugging Face's Transformer library for Qwen VL, Qwen Omni, and InternVL (Bai et al., 2025b;a; Xu et al., 2025; Wang et al., 2025). We adopt "qwen2.5_vl_7b", "qwen3_vl_2b", "qwen3_vl_4b", "qwen3_vl_8b", "qwen2.5_omni_7b". Additionally, we use "internvl35_8b".

# B. Experiments

## B.1. Modality Gap

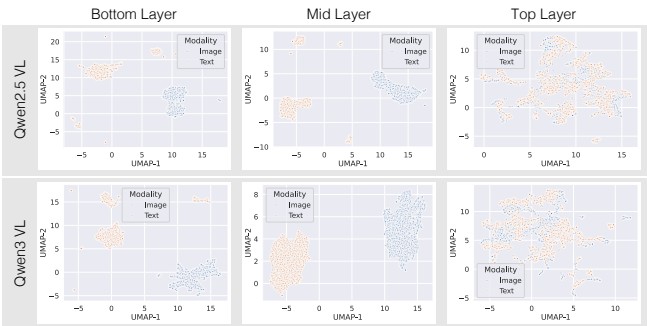

*Figure 6.* UMAP visualization of image and text embeddings of Qwen 2.5 VL 7B and Qwen3 VL 8B.

We use Qwen-VL (Bai et al., 2025b;a) and the COCO validation set (Lin et al., 2014). As mentioned in the main paper, we employ the prompts *"Summarize this image in one word:"* for image embeddings and *"Summarize this sentence in one word:"* for text embeddings. We show UMAP for modality gap following Liang et al. (2022). As shown in Fig. 6, the modality gap is small at the last layer.

**Logit-Lens JSD computation.** For the layer-wise Jensen–Shannon divergence used in the main paper, we apply the Logit Lens (nostalgebraist, 2020) to the last-token hidden state of each layer. Concretely, for a layer $l$ we take the last-token hidden state $\mathbf{h}_N^l$ and apply the model's final unembedding matrix followed by a softmax to obtain a probability distribution over the full vocabulary $\mathcal{V}$. We compute this distribution separately for the image input and the text input using the same instruction prompt, and report the Jensen–Shannon divergence between the two distributions at every layer. The divergence therefore compares image- and text-induced next-token distributions over the entire vocabulary $\mathcal{V}$ at each layer, without restricting to any subset of tokens.

## B.2. Other MLLMs

*Table 7.* Zero-Shot Rankability of InternVL 3.5 8B. We measure the Spearman rank correlation coefficient (SRCC).

| Model | UTKFace | Adience | QNRF | ST-A | ST-B | KonIQ | HCI |
|---|---|---|---|---|---|---|---|
| Qwen 2.5 VL 7B | 0.783 | 0.880 | 0.756 | 0.636 | 0.880 | 0.600 | 0.571 |
| Qwen 3.0 VL 8B | 0.804 | 0.888 | 0.747 | 0.509 | 0.875 | 0.613 | 0.465 |
| InternVL 3.5 8B | 0.772 | 0.847 | 0.725 | 0.496 | 0.881 | 0.769 | 0.350 |

In the main paper, we use Qwen Series for MLLM experiments because it is widely known as a strong open-source MLLM in the community. We additionally report InternVL 3.5 8B experiments for zero-shot rankability. See Table 7. InternVL is comparable to Qwen MLLMs.

## B.3. Specific Results

**UTKFace** See Table 8. Zero-shot rankability of MLLMs on UTKFace outperforms the zero-shot rankability of VLMs.

*Table 8.* UTKFace. We measure the Spearman rank correlation coefficient (SRCC).

| Embedding | | Model | Rankability | Extreme | Zero-Shot Rankability |
|---|---|---|---|---|---|
| VM | Vision-only | RN | 0.521 | 0.335 | - |
| | | ViTB32 | 0.754 | 0.648 | - |
| | | CNX | 0.785 | 0.645 | - |
| | | DINO-B14 | 0.768 | 0.632 | - |
| | VLM | CLIP-RN50 | 0.796 | 0.707 | 0.676 |
| | | CLIP-ViTB32 | 0.806 | 0.706 | 0.711 |
| | | CLIP-CNX | 0.810 | 0.763 | 0.771 |
| MLLM | | Qwen 2.5 VL 7B | 0.802 | 0.643 | 0.783 |
| | | Qwen 3.0 VL 2B | 0.816 | 0.658 | 0.800 |
| | | Qwen 3.0 VL 4B | 0.822 | 0.670 | 0.800 |
| | | Qwen 3.0 VL 8B | 0.824 | 0.686 | 0.804 |

**Adience**    See Table 9. Zero-shot rankability of MLLMs on Adience outperforms the zero-shot rankability of VLMs.

*Table 9.* Adience. We measure the Spearman rank correlation coefficient (SRCC).

| Embedding | | Model | Rankability | Extreme | Zero-Shot Rankability |
|---|---|---|---|---|---|
| VM | Vision-only | RN | 0.627 | 0.434 | - |
| | | ViTB32 | 0.874 | 0.837 | - |
| | | CNX | 0.879 | 0.830 | - |
| | | DINO-B14 | 0.888 | 0.829 | - |
| | VLM | CLIP-RN50 | 0.888 | 0.867 | 0.816 |
| | | CLIP-ViTB32 | 0.913 | 0.872 | 0.803 |
| | | CLIP-CNX | 0.911 | 0.870 | 0.791 |
| MLLM | | Qwen 2.5 VL 7B | 0.913 | 0.851 | 0.880 |
| | | Qwen 3.0 VL 2B | 0.924 | 0.855 | 0.908 |
| | | Qwen 3.0 VL 4B | 0.915 | 0.867 | 0.885 |
| | | Qwen 3.0 VL 8B | 0.923 | 0.860 | 0.888 |

**UCF-QNRF**    See Table 10. Zero-shot rankability of MLLMs on UCF-QNRF outperforms the zero-shot rankability of VLMs.

*Table 10.* UCF-QNRF. We measure the Spearman rank correlation coefficient (SRCC).

| Embedding | | Model | Rankability | Extreme | Zero-Shot Rankability |
|---|---|---|---|---|---|
| VM | Vision-only | RN | 0.712 | 0.573 | - |
| | | ViTB32 | 0.829 | 0.724 | - |
| | | CNX | 0.820 | 0.507 | - |
| | | DINO-B14 | 0.829 | 0.538 | - |
| | VLM | CLIP-RN50 | 0.880 | 0.564 | 0.585 |
| | | CLIP-ViTB32 | 0.869 | 0.558 | 0.592 |
| | | CLIP-CNX | 0.866 | 0.512 | 0.619 |
| MLLM | | Qwen 2.5 VL 7B | 0.858 | 0.425 | 0.756 |
| | | Qwen 3.0 VL 2B | 0.884 | 0.536 | 0.758 |
| | | Qwen 3.0 VL 4B | 0.895 | 0.588 | 0.812 |
| | | Qwen 3.0 VL 8B | 0.881 | 0.545 | 0.747 |

**ST-A**    See Table 11. Zero-shot rankability of MLLMs on ST-A outperforms the zero-shot rankability of VLMs except for Qwen 3.0 VL 8B.

*Table 11.* ST-A. We measure the Spearman rank correlation coefficient (SRCC).

| Embedding | | Model | Rankability | Extreme | Zero-Shot Rankability |
|---|---|---|---|---|---|
| VM | Vision-only | RN | 0.622 | 0.540 | - |
| | | ViTB32 | 0.764 | 0.756 | - |
| | | CNX | 0.717 | 0.659 | - |
| | | DINO-B14 | 0.696 | 0.579 | - |
| | VLM | CLIP-RN50 | 0.781 | 0.576 | 0.567 |
| | | CLIP-ViTB32 | 0.737 | 0.653 | 0.511 |
| | | CLIP-CNX | 0.775 | 0.512 | 0.607 |
| | MLLM | Qwen 2.5 VL 7B | 0.824 | 0.572 | 0.636 |
| | | Qwen 3.0 VL 2B | 0.814 | 0.614 | 0.675 |
| | | Qwen 3.0 VL 4B | 0.827 | 0.630 | 0.655 |
| | | Qwen 3.0 VL 8B | 0.837 | 0.658 | 0.509 |

**ST-B**    See Table 12. Zero-shot rankability of MLLMs on ST-B outperforms the zero-shot rankability of VLMs.

*Table 12.* ST-B. We measure the Spearman rank correlation coefficient (SRCC).

| Embedding | | Model | Rankability | Extreme | Zero-Shot Rankability |
|---|---|---|---|---|---|
| VM | Vision-only | RN | 0.572 | 0.549 | - |
| | | ViTB32 | 0.877 | 0.771 | - |
| | | CNX | 0.862 | 0.634 | - |
| | | DINO-B14 | 0.819 | 0.671 | - |
| | VLM | CLIP-RN50 | 0.894 | 0.739 | 0.567 |
| | | CLIP-ViTB32 | 0.854 | 0.757 | 0.670 |
| | | CLIP-CNX | 0.888 | 0.740 | 0.733 |
| | MLLM | Qwen 2.5 VL 7B | 0.916 | 0.738 | 0.880 |
| | | Qwen 3.0 VL 2B | 0.886 | 0.755 | 0.884 |
| | | Qwen 3.0 VL 4B | 0.915 | 0.797 | 0.928 |
| | | Qwen 3.0 VL 8B | 0.929 | 0.790 | 0.875 |

**AVA**    See Table 13. Zero-shot rankability of MLLMs on AVA outperforms the zero-shot rankability of VLMs.

*Table 13.* AVA. We measure the Spearman rank correlation coefficient (SRCC).

| Embedding | | Model | Rankability | Extreme | Zero-Shot Rankability |
|---|---|---|---|---|---|
| VM | Vision-only | RN | 0.465 | 0.120 | - |
| | | ViTB32 | 0.609 | 0.314 | - |
| | | CNX | 0.644 | 0.310 | - |
| | | DINO-B14 | 0.565 | 0.262 | - |
| | VLM | CLIP-RN50 | 0.701 | 0.433 | 0.463 |
| | | CLIP-ViTB32 | 0.705 | 0.466 | 0.486 |
| | | CLIP-CNX | 0.746 | 0.740 | 0.459 |
| | MLLM | Qwen 2.5 VL 7B | 0.682 | 0.303 | 0.600 |
| | | Qwen 3.0 VL 2B | 0.699 | 0.341 | 0.584 |
| | | Qwen 3.0 VL 4B | 0.694 | 0.360 | 0.597 |
| | | Qwen 3.0 VL 8B | 0.720 | 0.376 | 0.613 |

**KonIQ-10k**    See Table 14. Zero-shot rankability of MLLMs on KonIQ-10k outperforms the zero-shot rankability of VLMs.

*Table 14*. KonIQ-10k. We measure the Spearman rank correlation coefficient (SRCC).

| Embedding | | Model | Rankability | Extreme | Zero-Shot Rankability |
|---|---|---|---|---|---|
| VM | Vision-only | RN | 0.688 | 0.359 | - |
| | | ViTB32 | 0.737 | 0.552 | - |
| | | CNX | 0.749 | 0.346 | - |
| | | DINO-B14 | 0.736 | 0.592 | - |
| | VLM | CLIP-RN50 | 0.827 | 0.644 | 0.466 |
| | | CLIP-ViTB32 | 0.811 | 0.599 | 0.502 |
| | | CLIP-CNX | 0.875 | 0.729 | 0.508 |
| | MLLM | Qwen 2.5 VL 7B | 0.711 | 0.518 | 0.763 |
| | | Qwen 3.0 VL 2B | 0.795 | 0.616 | 0.768 |
| | | Qwen 3.0 VL 4B | 0.772 | 0.622 | 0.786 |
| | | Qwen 3.0 VL 8B | 0.794 | 0.606 | 0.778 |

**HCI**   See Table 15. Zero-shot rankability of Qwen 2.5 VL 7B on HCI outperforms VLMs. CLIP-RN50 and ViTB32 show strong zero-shot rankability compared to Qwen 3.0 VL series.

*Table 15*. HCI. We measure the Spearman rank correlation coefficient (SRCC).

| Embedding | | Model | Rankability | Extreme | Zero-Shot Rankability |
|---|---|---|---|---|---|
| VM | Vision-only | RN | 0.466 | 0.321 | - |
| | | ViTB32 | 0.530 | 0.370 | - |
| | | CNX | 0.621 | 0.491 | - |
| | | DINO-B14 | 0.476 | 0.486 | - |
| | VLM | CLIP-RN50 | 0.759 | 0.546 | 0.533 |
| | | CLIP-ViTB32 | 0.761 | 0.553 | 0.503 |
| | | CLIP-CNX | 0.812 | 0.603 | 0.362 |
| | MLLM | Qwen 2.5 VL 7B | 0.683 | 0.292 | 0.571 |
| | | Qwen 3.0 VL 2B | 0.676 | 0.344 | 0.418 |
| | | Qwen 3.0 VL 4B | 0.692 | 0.354 | 0.484 |
| | | Qwen 3.0 VL 8B | 0.701 | 0.364 | 0.465 |

