# OpenReview forum: "Zero-Shot Rankability: Revealing Latent Ordinal Structure in Multimodal Large Language Models via Language"
_ICML.cc/2026/Conference — ICML 2026 regular_

### Official Review · Reviewer_khp6 · 2026-03-01

**Soundness:** 2
**Presentation:** 2
**Significance:** 2
**Originality:** 3
**Overall Recommendation:** 4
**Confidence:** 2

**Summary:**

This paper investigates the latent ordinal structure within Multimodal Large Language Models (MLLMs), specifically focusing on the zero-shot recovery of ranking axes (e.g., age, crowd count, aesthetics). Building upon recent work that identified "rankability" in vision encoders, the authors address the sub-optimality of zero-shot text-driven rank axes in traditional Vision-Language Models (VLMs) like CLIP. The authors propose a simple baseline that leverages the generative nature of MLLMs: extracting embeddings from the last token using conditional prompts (e.g., "Describe the color...").

The study demonstrates that MLLMs can recover approximately 90% of the performance of a supervised linear probe in a zero-shot manner, significantly outperforming VLMs which only recover about 61%. The authors attribute this success to two primary factors: the ability of conditional prompts to disentangle polysemantic features, and a reduced modality gap in the deeper layers of MLLMs. Finally, the work extends this analysis to the audio modality using Omni-models, showing effective cross-modal transfer from Audio-to-Vision.

**Compliance With Llm Reviewing Policy:**

Affirmed.

**Final Justification:**

I raise my score from 3 to 4, as I think the rebuttal solved my concerns. But I lower my confidence from 3 to 2, as I find other reviewers obviously understand this field better than I do.

**Key Questions For Authors:**

1.  Your main results focus on the Qwen series. Do these findings hold for MLLM architectures that do not feed visual tokens directly into the LLM via a projector (e.g., architectures using cross-attention mechanisms like IDEFICS or Flamingo variants)? Does the "conditional embedding" hypothesis hold when the visual encoder is frozen and distinct?
2.   How sensitive is the derived rank axis $a_c$ to the specific choice of antonyms in the text prompts? For example, does changing "aged person" vs. "young person" to "elderly" vs. "child" significantly alter the projection direction?
3.   In Figure 4, you attribute improved rankability to the reduced modality gap. However, deeper layers also generally contain higher-level semantic abstractions. Have you performed any experiments (perhaps with models that have a large modality gap but high reasoning capability) to isolate the effect of the modality gap from general semantic capability?
4.   In Table 6, Vision-to-Audio transfer performs significantly worse than Audio-to-Vision. You attribute this to the "difficulty" of the task. Could this instead be due to the asymmetry in the training data volume (text-image vs. text-audio) leading to a less aligned geometry for audio in the shared space?

**Limitations:**

yes

**Strengths And Weaknesses:**

**Strengths:**
*   Understanding the internal geometry of Large Multimodal Models beyond simple semantic similarity is crucial for interpretability and control. The focus on ordinality—continuous attributes rather than categorical classification—provides valuable insights into how these models structure the world.
*   This submission intends to focus on a broad topic regarding the geometry of embeddings, but offers a specific, novel contribution by shifting the focus from VLMs to MLLMs. The connection drawn between "conditional embeddings" and the reduction of interference from polysemantic features (Equation 1) is a compelling theoretical intuition that explains the empirical success.

**Weaknesses:**
*   The main body of the paper relies heavily on the Qwen-VL series. While InternVL is mentioned in the Appendix, the architectural differences between MLLMs (e.g., those using cross-attention like Flamingo vs. those using projection layers like LLaVA/Qwen) might yield different embedding behaviors. The generalization of these findings to other architectural families is not fully explored.
*   The paper argues that a small modality gap enables better rankability. While the correlation in Figure 4 is strong (deeper layers = smaller gap = better SRCC), the causal link is not definitively proven. It is possible that deeper layers simply possess better semantic abstraction, which improves ranking regardless of the modality gap distance.
*    The method relies on constructing a text axis $a_c = h_{pos} - h_{neg}$. There is no analysis of how sensitive the rank axis is to the specific phrasing of the positive and negative prompts. Given that LLMs are known to be sensitive to prompting, this is a missing ablation.
*   Using an MLLM (even a 7B model) to extract embeddings for a ranking task is significantly more computationally expensive than using a lightweight CLIP encoder. The paper does not discuss the trade-off between the performance gain (90% recovery vs. 60%) and the inference latency/cost.

---

> ### Author Rebuttal · Authors · 2026-03-31
>
> We thank the reviewer for the constructive and detailed feedback.
>
> **Key Question 1 & Weakness 1\. Ablation study of cross-attention mechanisms**
>
> We report results for IDEFICS1 (cross-attention-based, 9B) and IDEFICS2 (projector-based, 8B). On UTKFace, IDEFICS1 achieves an SRCC of 0.774, while IDEFICS2 reaches 0.751. For KonIQ, IDEFICS1 attains 0.338, whereas IDEFICS2 attains 0.718, which aligns with both Qwen and InternVL. The particularly low zero-shot rankability of IDEFICS1 on KonIQ suggests reduced effectiveness in cross-attention-based architectures.
>
> *Among MLLM designs, projector-based architectures have become widespread. Consequently, we view the geometry of these models as a highly relevant area of study. Consistency in* our findings across Qwen, InternVL, and IDEFICS2 indicates the observed phenomenon likely generalizes to other representative projector-based MLLMs.
>
> **Key Question 2 & Weakness 3\. Prompt Sensitivity**
>
> *MLLMs reveal more ordinal geometric structure than VLMs across diverse prompt choices.* Our prompt search follows the established protocol of Sonthalia et al. (2025), operating only on the validation set with test sets remaining completely unseen. For prompt sensitivity, Qwen 2.5 VL 7B achieves a mean SRCC of 0.728 ± 0.061 on UTKFace and 0.409 ± 0.182 on KonIQ across all prompt pairs, compared to CLIP-CNX's 0.555 ± 0.168 and 0.157 ± 0.170. This is consistent with our main finding.
>
> **Key Question 3 & Weakness 2\. Modality Gap**
>
> *Reducing the modality gap itself improves zero-shot rankability.* Finding an MLLM with both a large modality gap and high reasoning ability is not straightforward. As an alternative, we provide a controlled experiment by applying a post-hoc modality gap reduction method (I0T: Embedding Standardization Method Towards Zero Modality Gap, ACL 2025\) to VLMs. This approach reduces the modality gap while keeping the architecture and semantic capability unchanged.
>
> On KonIQ, where the gap between rankability and zero-shot rankability is particularly large, SRCC improves from 0.499 to 0.661 (CLIP-RN50), 0.502 to 0.663 (CLIP-ViTB32), and 0.508 to 0.738 (CLIP-CNX). This controlled experiment suggests that reducing the modality gap itself contributes to improved zero-shot rankability.
>
> **Key Question 4\. Training data volume (text-image vs. text-audio)**
>
> We truly appreciate the reviewer’s insightful observation. It is worth noting that data volume and information volume are intricately linked—voice signals naturally contain less age-discriminative information than facial images, leading to less aligned geometry in the audio embedding space. The impact of information volume is effectively illustrated by the much lower upper bound on rankability for audio (0.468) compared to vision (about 0.8). We look forward to discussing this further in our revised manuscript.
>
> **Weakness 4\. Inference Cost**
>
> For reference, on an NVIDIA RTX A6000, CLIP-CNX (352M params) runs at 13.92 ms/img, while Qwen 3.0 VL 2B (2B params) runs at 52.76 ms/img.
>
> We wish to clarify our primary contribution: an empirical and geometric analysis of ordinal structure in MLLM embedding spaces. The computational trade-off, therefore, stands orthogonal to our main claims. Furthermore, as the use of MLLM embeddings for downstream tasks is an actively emerging research area, understanding their geometric properties represents a necessary groundwork for broader adoption.

---

> > ### Author Rebuttal · Reviewer_khp6 · 2026-04-02
> >
> > I raise my score from 3 to 4, as I think the rebuttal solved my concerns. But I lower my confidence from 3 to 2, as I find other reviewers obviously understand this field better than I do.

---

> > > ### Author Response · Authors · 2026-04-08
> > >
> > > We thank the reviewer for reconsidering the score and for the constructive feedback. We are glad that the rebuttal addressed the concerns, and we will incorporate the discussed questions into the final manuscript.

---

### Official Review · Reviewer_2ztX · 2026-03-13

**Soundness:** 2
**Presentation:** 3
**Significance:** 2
**Originality:** 3
**Overall Recommendation:** 3
**Confidence:** 2

**Summary:**

This paper studies the rankability of embeddings in multimodal large language models (MLLMs). The authors show that ordinal attributes (e.g., age, crowd count, aesthetics) can be recovered in a zero-shot manner by constructing a ranking axis from antonym text prompts. Experiments on several datasets suggest that MLLM embeddings achieve strong zero-shot ranking performance, approaching the performance of supervised linear probing.

**Compliance With Llm Reviewing Policy:**

Affirmed.

**Key Questions For Authors:**

See Weaknesses.

**Limitations:**

yes

**Strengths And Weaknesses:**

Strengths:
- The paper studies ordinal structure in MLLM embeddings, which is a relatively underexplored angle. The idea of probing ranking directions through language prompts is simple but conceptually interesting.

Weaknesses:
- The proposed method is extremely simple, essentially computing a difference between two text embeddings to define a ranking axis. This makes the contribution feel more like an empirical observation than a new technical method.

- It is not entirely clear how this approach would compare to simply prompting the model to perform pairwise comparisons directly. The paper could better clarify the scenarios where this embedding-based ranking approach is preferable.

---

> ### Author Rebuttal · Authors · 2026-03-31
>
> We thank the reviewer for the constructive feedback.
>
> **Weakness 1\. Empirical observation rather than a new technical method.**
>
> *This work makes a knowledge, not technical, contribution. We believe this is both valid and valuable. The technically simple investigation approach reveals a property of the model; such simplicity is not a weakness*.
>
> This framing aligns with prior work (Sonthalia et al., 2025), which showed that modern visual embeddings have ordinal geometry that is recoverable via supervised linear probing. It also showed that zero-shot axis recovery in VLMs is suboptimal. Our paper extends this line of investigation to MLLMs. We ask whether the same suboptimality persists and find that it does not when MLLMs' distinct characteristics (conditional embeddings and a smaller modality gap) are leveraged.
>
> The internal geometry of MLLM embeddings is underexplored, making such empirical findings valuable. Demonstrating and explaining zero-shot rankability directly advances understanding of MLLM embeddings.
>
> **Weakness 2\.  Comparison with prompting for pairwise comparisons directly.**
>
> The two approaches offer different capabilities. Pairwise prompting gives only a relative order between two items and cannot produce a continuous scalar value for attribute intensity. *In contrast, an embedding with ordinal geometry gives a continuous projection score ($s = \mathbf{a}^⊤\mathbf{h}$). This score encodes both the ordinal meaning and the intensity differences between items. By 'interval information,' we mean the measurement of how much one item's attribute differs from another's, not just their order.* As a result, our method estimates both the order and the degree of separation, which pairwise comparisons cannot. It also enables direct use in regression tasks and in applications that need absolute magnitude, not just relative order.
>
> *Generation-based ranking requires O(N²) or O(N log N) MLLM inference calls. Our embedding approach needs only O(N) calls, one forward pass per image.* Adding a new image with generation-based methods requires O(N) more inference calls. Our method only needs one forward pass and one vector similarity operation.
>
> For the reviewer interested in a generation-based approach that outputs continuous attribute values directly via text, we refer to our response to Reviewer XpQi (Key Question 4).

---

> > ### Author Rebuttal · Reviewer_2ztX · 2026-04-03
> >
> > The authors have partially addressed my concerns about the simplicity of the method and the comparison with pairwise approaches.
> > I still have some doubts about the technical novelty, and therefore I maintain my score.

---

> > > ### Author Response · Authors · 2026-04-04
> > >
> > > We appreciate the reviewer's acknowledgment of the method's simplicity and the comparison with pairwise approaches. We would like to reiterate that our technical novelty does not lie in the construction of a ranking vector itself, but in uncovering why MLLM embeddings succeed where VLM embeddings fail.
> > >
> > > This required recognizing that MLLMs possess two distinct properties that overcome the limitations of VLMs: (1) conditional embeddings, enabled by MLLMs' generative nature, disentangle polysemantic features and allow a text-conditioned rank axis to transfer effectively to the visual modality; and (2) a smaller vision-text modality gap further enables this transfer. These insights are non-trivial: they required systematic analysis across modalities and architectures.
> > >
> > > The value of the ranking approach lies not merely in its empirical performance, but in the fact that our findings both support the method and explain why it works. Without this understanding, drawing a text-driven line in embedding space could look like an ad hoc choice that merely happens to perform well. With it, the approach becomes a finding-driven method whose value lies not only in its effectiveness but also in the insight it provides.
> > >
> > > If the reviewer's remaining doubts go beyond what we addressed above, we would appreciate further specification, as we are eager to address any additional concerns directly.

---

### Official Review · Reviewer_HQhb · 2026-03-15

**Soundness:** 2
**Presentation:** 2
**Significance:** 2
**Originality:** 2
**Overall Recommendation:** 3
**Confidence:** 4

**Summary:**

This paper studies whether MLLMs possess a latent, approximately linear ordinal structure that can be accessed in a zero-shot manner via language alone.

**Compliance With Llm Reviewing Policy:**

Affirmed.

**Key Questions For Authors:**

How exactly are the “100 pairs of texts and 100 image prompts” used in the zero-shot setting? Do you select the best-performing pair post hoc, average over pairs, or ensemble? Please clarify construction, selection, and whether any validation tuning on test data occurred.
What are the precise prompts (T_c^pos, T_c^neg, P_c) used per attribute and dataset? Could you provide a complete prompt list and report sensitivity (mean±std) across paraphrases/antonyms?
For the VLM baselines, did you attempt any form of prompt ensembling or text-anchor averaging analogous to the 100-pair setting to ensure a fair comparison? If not, can you add this control or discuss why it is infeasible?
How did you control for possible training-data leakage from widely used datasets (UTKFace, Adience, AVA, HCI) into the pretraining corpora of the MLLMs? Any near-duplicate filtering or sensitivity analyses?
Please detail the logit-lens JSD computation: which distributions are compared, over what token sets, and how are image hidden states mapped to logits? Are numbers (not only trends) consistent across random seeds?
In Table 6 (cross-modal transfer), if the axis is text-derived, why does the source dataset affect the axis? Are dataset-conditioned or modality-conditioned steps used to estimate the axis beyond the two text anchors?
For audio (Common Voice), what prompts and conditioning were used to obtain audio embeddings? Are the age labels categorical or continuous, and how is SRCC computed in that setting?
Could you report per-layer angular distances between the linear-probed visual axis a_I and the text-defined axis a_L to directly test the “modality-gap alignment” hypothesis?
How many samples and what split were used to train the linear probes (“upper bound”) per dataset? Is the probe trained strictly on train and evaluated on held-out test with fixed hyperparameters or via CV?

**Limitations:**

As mentioned above.

**Strengths And Weaknesses:**

The paper introduces and empirically validates the notion of zero-shot rankability for MLLM embeddings, showing that a simple, text-only defined rank axis can be effective without labelled data.

However, the comparison to VLMs is potentially confounded: MLLMs leverage conditional embeddings for images (instruction-conditioned hidden states), while CLIP-style VLMs cannot produce image embeddings conditioned in the same way. Some of the measured advantages could reflect access to conditional representations rather than an inherently better cross-modal rank axis.

The modality-gap evidence (UMAP visuals and a logit-lens JSD trend) is primarily qualitative/indicative rather than a rigorous, controlled quantification; alternative, more direct alignment measures or controls would strengthen the claim.

The audio extension and cross-modal transfer setup are under-specified methodologically, making it hard to assess whether the text-defined rank axis is truly modality-agnostic or whether additional modality-specific operations were used.

---

> ### Author Rebuttal · Authors · 2026-03-31
>
> We thank the reviewer for the careful and detailed reading.
>
> **Key Question 10\. Details of linear probes.**
>
> *Linear rankability and the linear probing protocol are contributions of Sonthalia et al. (2025) \[A1\], not to us. We adopt* their implementation to ensure consistency with the established upper bound. Training, validation, and test sets are strictly separated. Hyperparameters are selected using only validation, and the test set is evaluated only once. If there is no official validation split, 10% of the training data is used for validation, with a fixed random seed for reproducibility. For instance, UTKFace comprises 13,146 training/validation samples and 3,287 test samples. We will include more details in the revised manuscript.
>
> \[A1\] On the rankability of visual embeddings, NeurIPS 2025
>
> **Key Question 1\. Details of Prompt Search in the zero-shot setting.**
>
> *The 100 prompt pairs are selected from the validation set only, ensuring no test data leakage.* For further details, refer to our response to Reviewer XpQi (Key Question 1 and Limitation 1\) due to space limits.
>
> **Key Question 2\. Details of Prompt.**
>
> As a representative example, we provide the selected prompts for the Age attribute:
>
> * $T\_c^{pos}$: A full-body shot of a man in his late 70s.
> * $T\_c^{neg}$: A full-body shot of a boy in his early teens.
> * $P\_c$: Life stage (one word): *The full prompt list will be released due to space limits.*
>
> For prompt sensitivity, Qwen2.5 VL 7B achieves a mean SRCC of 0.728±0.061 on UTKFace and 0.409±0.182 on KonIQ, consistently outperforming CLIP-CNX at 0.555±0.168 and 0.157±0.170, respectively. These results support our main finding that MLLMs demonstrate greater *ordinal geometric structure in zero-shot settings than VLMs.*
>
> **Key Question 3\. Fair comparison for VLM baselines.**
>
> *We used the same 100-pair prompt search protocol for the VLM baseline \[A1\]. This confirms our evaluation does not disadvantage VLMs.* Table 2 shows that our reproduction (0.497) is comparable to, or slightly higher than, the original (0.485). We also tried prompt ensembling on KonIQ (SRCC 0.370), confirming best-pair selection favors VLMs over ensembling.
>
> **Key Question 4\. Data Leakage.**
>
> Due to the space limitation, we kindly refer the reviewer to our response to Reviewer XpQi (Key Question 2 and Limitation 2).​
>
> **Key Question 5\. Details of the logit-lens JSD computation.**
>
> Applying the logit-lens approach, we project the last token's hidden states from each layer into the vocabulary space using the final unembedding matrix, yielding a probability distribution over the full vocabulary. We then compute the JSD between these two distributions for each layer.
>
> **Key Question 6\. In Table 6 (cross-modal transfer), why does the source dataset affect the axis?**
>
> *Dataset-dependent variation is expected, as the optimal prompt pair is selected for each source dataset using its validation set.* This aligns with previous findings indicating that even linear axes for the same attribute may differ across datasets.
>
> **Key Question 7\. Details of Cross-Modal Experiment.**
>
> GPT-generated prompts are used following \[A1\]. Instead of dataset- or modality-based steps, we condition on the attribute, such as age or aesthetics. For audio, we use the prompt: "Identify the age group of this person in one word."
>
> **Key Question 8\. Are the age labels categorical or continuous, and how is SRCC computed in that setting?**
>
> Age labels are categorical in Common Voice and Adience, and continuous in UTKFace. SRCC is appropriate for both types, as it quantifies correlation by comparing ranks rather than raw values.
>
> **Key Question 9\. Test modality-gap alignment hypothesis**
>
> We conducted a controlled experiment applying a post-hoc modality gap reduction method \[A2\] to VLMs. On KonIQ, SRCC increased from 0.466 to 0.661 (CLIP-RN50), 0.502 to 0.663 (CLIP-ViTB32), and 0.508 to 0.738 (CLIP-CNX). *These results support the hypothesis that mitigating the modality gap enhances zero-shot rankability.*
>
> \[A2\] 10T: Embedding … Zero Modality Gap, ACL 2025
>
> ​**Weakness 1\. Conditional embeddings and fairness of comparison.**
>
> *Conditional embedding is not a confound; it is a key feature of MLLM embeddings.* We begin from the established finding that zero-shot rankability is suboptimal for VLM embeddings (Sonthalia et al., 2025). Our main question: do MLLM embeddings, with fundamentally different characteristics, exhibit zero-shot rankability? We do not aim for a direct comparison between VLMs and MLLMs. Understanding how conditional embedding enables this is central **to our work**.

---

### Official Review · Reviewer_XpQi · 2026-03-16

**Soundness:** 3
**Presentation:** 3
**Significance:** 3
**Originality:** 3
**Overall Recommendation:** 3
**Confidence:** 4

**Summary:**

This paper investigates whether Multimodal Large Language Models (MLLMs) encode ordinal structure in their embedding spaces without any rank-specific training. The authors show that MLLMs achieve an average Spearman Rank Correlation (SRCC) of 0.728 in zero-shot ranking tasks, recovering ~90% of the supervised upper bound — far exceeding VLMs at 0.497. The key mechanisms identified are: (1) conditional embeddings via text prompts that define the ranking axis, and (2) a smaller modality gap in MLLMs compared to VLMs. The method uses antonymous prompts (e.g., "young"/"old") to derive a rank axis in embedding space. Evaluated on 8 vision datasets (Age, Crowd density, Aesthetics, Recency, etc.) with extension to audio modality.

**Compliance With Llm Reviewing Policy:**

Affirmed.

**Key Questions For Authors:**

1. How sensitive are the SRCC results to the choice of antonymous prompt pair? Have you tried multiple prompt pairs per concept and measured variance?
2. Can you rule out training data contamination as the primary explanation? Are any of your evaluation datasets known to appear in common pretraining corpora?
3. What happens when you apply the antonymous prompt technique to VLMs with artificially reduced modality gaps (e.g., via linear projection)? Does ranking improve?
4. How does embedding-based ranking compare to asking the MLLM to explicitly rank items via text generation?
5. Per-dataset breakdown: which ranking concepts show the largest gap to supervised performance, and why?

**Limitations:**

**"Zero-shot" is somewhat misleading.**

The method requires choosing the right antonymous prompt pair, which implicitly encodes human knowledge about the ranking concept. Finding the right prompt pair is itself a form of supervision/tuning. How sensitive are results to prompt choice? What if the antonym pair is suboptimal?

**Confound: training data contamination.**

MLLMs are trained on massive internet data that includes rankings (product ratings, age estimation, beauty contests). The "zero-shot" ordinal structure may simply reflect memorized ranking patterns from pretraining data, not emergent geometric structure. This alternative explanation is insufficiently addressed.

**Limited concept space.**

The 8 datasets cover relatively "easy" ranking concepts where language has clear ordinal descriptors. What about abstract or domain-specific ranking tasks (e.g., geological age of rocks, severity of medical conditions, complexity of algorithms)? The method likely fails when language doesn't naturally capture the ordinal axis.

**Strengths And Weaknesses:**

**Genuinely surprising finding.**

That MLLMs can rank without any ranking supervision is a notable empirical discovery with implications for understanding what these models learn.

**Mechanistic analysis is compelling.**

The modality gap analysis and conditional embedding explanation go beyond just reporting numbers — they provide interpretable reasons for the observed behavior.

**Elegant method.**

The antonymous prompt approach is simple, intuitive, and requires no training — a nice contribution to the "probing" literature for multimodal models.

**Breadth of evaluation.**

8 diverse datasets spanning different ranking concepts (age, aesthetics, crowd density, temporal recency) demonstrate generality.

---

> ### Author Rebuttal · Authors · 2026-03-31
>
> We thank the reviewer for the thorough and insightful review.
>
> **Key Question 1 and Limitation 1: Prompt Sensitivity and "Zero-shot" Terminology**
>
> **Q1-1 Zero-shot terminology**
>
> *We follow the terminology and protocol of \[A1\], who conducted a prompt search.* This prompt search uses the validation set only and involves no labeled training data. By the standard definition, it remains zero-shot. However, 'zero-shot' can mean different things. We will clarify in the manuscript that our prompt search used a validation set. This separates our approach from **True zero-shot** **research** \[A2\], which does not use a development set for few- or zero-shot tasks.
>
> \[A1\] On the rankability of visual embeddings, NeurIPS 2025
>
> \[A2\] A Simple Zero-shot Prompt ..., ICML 2024
>
> **Q1-2 Sensitivity of Prompt**
>
> *MLLMs consistently reveal greater ordinal geometric structure than VLMs across all prompt choices.* We report mean ± standard deviation SRCC across all prompt pairs, without validation-based selection. Qwen2.5 VL 7B achieves a mean SRCC of 0.728 ± 0.061 on UTKFace and 0.409 ± 0.182 on KonIQ. CLIP-CNX achieves corresponding values of 0.555 ± 0.168 and 0.157 ± 0.170. These results align with our primary finding.
>
> For clarity, representative prompts are: $T\_c^{pos}$: *A full-body shot of a man in his late 70s,* $T\_c^{neg}$: *A full-body shot of a boy in his early teens,* $P\_c$: *Life stage (one word).*
>
> **Key Question 2 and Limitation 2: Data Contamination**
>
> *We agree that contamination cannot be ruled out entirely. However, it alone is unlikely to explain the observed effect.* Both VLMs and MLLMs are trained on large web-scale corpora and may have seen related data during pretraining. As shown in Fig. 1, despite similarly large-scale pretraining, VLMs and MLLMs have different multimodal embedding structures. VLMs retain a clear modality gap. In contrast, MLLMs form a much more aligned shared space.
>
> *The modality gap, not data contamination, determines cross-modal transfer.* When the modality gap is large, text-defined ordinal axes and image embeddings exist in different subspaces. This difference makes transfer between modalities ineffective. When image and text embeddings share the same space, a text-defined ordinal axis applies directly to image embeddings. This suggests zero-shot rankability is driven by the geometric structure of multimodal representations, not just data contamination.
>
> **Key Question 3: Reduced Modality Gap Technique To VLMs.**
>
> *Reducing the modality gap in VLMs improves zero-shot rankability. This result is consistent with our hypothesis.* We evaluate VLMs with a reduced modality gap using a post-hoc method \[A3\]. On KonIQ, where the gap between rankability and zero-shot rankability is large, SRCC improves from 0.499 to 0.687 across three CLIP variants. This supports our observation that a smaller modality gap is linked to higher zero-shot rankability.
>
> \[A3\] 10T: Embedding … Zero Modality Gap, ACL 2025
>
> **Key Question Q4: Comparison with text generation of MLLM**
>
> Our focus is not on direct comparison with generation-based methods, but on understanding whether ordinal structure is inherently preserved in MLLM embeddings.
>
> *Comparable performance provides evidence that ordinal structure is geometrically organized in MLLM embeddings.* Specifically, text-generation-based scoring generates text descriptions for inputs and evaluates these, while embedding-based ranking determines ordinal relationships by comparing input embeddings. Both protocols achieve strong results on UTKFace (0.772 vs. 0.783; Qwen 2.5 VL 7B). It further supports **our motivation section.**
>
> *In generation-based ranking, ranking is performed via autoregressive text generation, which is not practical for large datasets due to its high computational cost (152–1,600 ms vs. 92 ms for MLLM embedding extraction).* Embedding-based ranking uses only O(N) forward passes, with embeddings that can be reused for ranking, retrieval, and regression tasks, unlike generation-based ranking.
>
> For the pairwise-comparison approach, we refer the reviewer to our response to Reviewer 2ztX (Weakness 2).
>
> **Key Question 5 and Limitation 3: Per-dataset Analysis and Concept Coverage**
>
> *Our results are consistent with Sonthalia et al. (2025) \[A1\], who observed that certain attributes are harder to rank.* The eight datasets used in our paper follow the same protocol as \[A1\]. \[A1\] notes that "some attributes are better ranked than others" and hypothesizes that "attribute-wise rankabilities are directly proportional to attribute-wise variety present in the training data."
>
> *The largest gap appears in HCI, where the ordinal concept is hard to verbalize.* The decade of photography is difficult to express with antonyms because decade-specific visual traits are subtle. As the reviewer notes, some ordinal concepts are harder to convey when their visual traits are tough to describe in language. We acknowledge this as a limitation.

---

### Decision · Program_Chairs · 2026-04-30

**Decision:**

Accept (regular)

**Comment:**

This paper received mixed ratings: one Weak Accept (4) and three Weak Rejects (3). AC decided to downweight the review of Reviewer XpQi (Weak Reject), as they did not acknowledge the rebuttal or participate in the discussion phase.

After a personal assessment of both the manuscript and the rebuttal, AC determined that the work offers significant value to the ICML community. While some reviewers noted the method's technical simplicity, the AC believes the strength of the empirical discoveries outweighs these concerns and recommends a Weak Accept.

The main arguments of weak acceptance are:

- Significant empirical discovery. The paper reveals that MLLM embeddings, unlike those in VLM (CLIP), contain a latent ordinal structure. This structure can be recovered using simple, text-driven antonymous prompts, achieving nearly 90% of the supervised upper-bound performance. Reviewers highlighted this as a “genuinely surprising” and “notable” finding that advances our understanding of model learning.

- Insightful explanations. The authors identify conditional embeddings and a reduced modality gap as the primary drivers of this phenomenon. Reviewers found this analysis both "compelling" and "interpretable," noting that the connection to disentangling polysemantic features provides strong theoretical intuition.

- Effective rebuttal. The authors' response successfully addressed the majority of the concerns. Regarding the critique of "technical simplicity," the authors successfully argued that the value of the work lies in the significant discovery of zero-shot rankability in MLLMs and the explanations.

The AC encourages the authors to incorporate the following feedback from reviewers for the final version:

- Terminology: Clarify the use of "zero-shot" to avoid ambiguity.

- Reproducibility: Provide the full list of prompts used in the study and release the code.

- New results: Integrate the experiments and analyses provided during the rebuttal, specifically the modality gap reduction results and the prompt-sensitivity analysis.